# Seasonal Variations in Stroke and a Comparison of the Predictors of Unfavorable Outcomes among Patients with Acute Ischemic Stroke and Cardioembolic Stroke

**DOI:** 10.3390/biomedicines12010223

**Published:** 2024-01-19

**Authors:** Pei-Ya Chen, Wan-Ling Chang, Cheng-Lun Hsiao, Shinn-Kuang Lin

**Affiliations:** 1Stroke Center, Department of Neurology, Taipei Tzu Chi Hospital, Buddhist Tzu Chi Medical Foundation, New Taipei City 23142, Taiwan; ruentw@gmail.com (P.-Y.C.); laetitia0717@hotmail.com (W.-L.C.); shb@ms19.hinet.net (C.-L.H.); 2School of Medicine, Tzu Chi University, Hualien 97004, Taiwan

**Keywords:** acute ischemic stroke, atrial fibrillation, cardioembolic stroke, mortality, seasonal variation, unfavorable outcomes

## Abstract

We investigated the seasonal variations in stroke in 4040 retrospectively enrolled patients with acute ischemic stroke (AIS) admitted between January 2011 and December 2022, particularly those with cardioembolic (CE) stroke, and compared predictors of unfavorable outcomes between AIS patients and CE stroke patients. The classification of stroke subtypes was based on the Trial of ORG 10172 in Acute Stroke Treatment. Stroke occurrence was stratified by seasons and weekdays or holidays. Of all AIS cases, 18% were of CE stroke. Of all five ischemic stroke subtypes, CE stroke patients were the oldest; received the most thrombolysis and thrombectomy; had the highest initial National Institutes of Stroke Scale (NIHSS) and discharge modified Rankin Scale (mRS) scores; and had the highest rate of in-hospital complications, unfavorable outcomes (mRS > 2), and mortality. The highest CE stroke prevalence was noted in patients aged ≥ 85 years (30.9%); moreover, CE stroke prevalence increased from 14.9% in summer to 23.0% in winter. The main predictors of death in patients with CE stroke were age > 86 years, heart rate > 79 beats/min, initial NIHSS score > 16, neutrophil-to-lymphocyte ratio (NLR) > 6.4, glucose > 159 mg/dL, cancer history, in-hospital complications, and neurological deterioration (ND). The three most dominant factors influencing death, noted in not only patients with AIS but also those with CE stroke, are high initial NIHSS score, ND, and high NLR. We selected the most significant factors to establish nomograms for predicting fatal outcomes. Effective heart rhythm monitoring, particularly in older patients and during winter, may help develop stroke prevention strategies and facilitate early AF detection.

## 1. Introduction

Ischemic stroke accounts for more than 80% of all strokes worldwide. Ischemic stroke etiologies are mainly classified using the Trial of ORG 10172 in the Acute Stroke Treatment (TOAST) classification as follows: small-artery occlusion (SAO), large-artery atherosclerosis (LAA), cardioembolism, other determined etiology, and undetermined etiology [1]. The distribution of TOAST stroke subtypes differs by ethnicity [2]. For instance, SAO is the most common subtype in many Asian countries [3]. Cardioembolic (CE) stroke prevalence varies among different areas; nevertheless, it accounts for approximately 20% of all ischemic stroke cases worldwide and is continuing to increase [4]. However, a multicenter registry study reported that CE stroke cases constituted only 10.9% of all ischemic stroke cases in Taiwan [5]. CE stroke prevalence appears to be increasing because newer techniques have enabled the accurate categorization of patients with atrial fibrillation (AF) or other cardiac disorders, including CE stroke patients who were initially categorized as having a stroke due to an undetermined source [6]. Even though the prevalence of CE remains lower than that of SAO and LAA stroke, patients with CE stroke frequently present with higher initial stroke severity and poorer functional outcomes [4].

The leading cause of cardioembolic stroke is atrial fibrillation. CE stroke incidence is higher in the older population than in the general population. Nevertheless, compared with SAO and LAA stroke patients, those with CE stroke may have fewer comorbidities that lead to arteriosclerosis. Therefore, risk factors for poor prognosis and effective treatment methods in patients with CE stroke may differ from those in patients with other ischemic stroke subtypes. Previous single-center studies on the impact of seasonality on stroke incidence are conflicting, but most have reported an increase in stroke incidence and mortality during the cold winter months. However, recent meta-analytic studies of the effects of ambient temperature on stroke incidence have yielded inconsistent results. Lien et al. found that short-term changes in temperature (both low and high temperatures) had significant impacts on major adverse cerebrovascular events [7]. Li et al. found very little seasonal variation in an overall analysis of 33 studies [8]. Kuzmenko et al. found that the seasonal dynamics of ischemic stroke are not clearly expressed but are determined by regional climate characteristics [9]. It is possible that the influence of seasonal fluctuation on stroke development could be different based on the TOAST subtypes. Specific analyses of the impact of temperature on AF have revealed that the risk of AF is higher during the winter months [10]. In addition to its effect on blood pressure, temperature affects heart function and rhythm; therefore, it possibly contributes to CE stroke occurrence [11,12].

Understanding the relationship between regional climate differences and stroke could help develop stroke prevention strategies. In this study, we investigated seasonal and weekly changes in patients with acute ischemic stroke (AIS), including patients with CE stroke subtypes, and compared unfavorable outcome-related factors between all AIS patients and CE stroke patients in particular.

## 2. Materials and Methods

### 2.1. Study Population and Data Collection

The index hospital is located in the subtropical region of Northern Taiwan. For the continuous quality control of stroke care, we prospectively registered patients admitted for stroke to the appropriate ward. We retrospectively reviewed the records of all registered inpatients who presented with AIS from January 2011 to December 2022. AIS diagnosis was confirmed on the basis of clinical presentations and either the presence of an ischemic lesion or the absence of a corresponding intracranial lesion other than infarction on brain computed tomography or magnetic resonance images. The following information was collected: age, sex, and clinical features, including initial presentation, initial National Institutes of Health Stroke Scale (NIHSS) score, stroke risk factors, and hospital length of stay (LOS). Laboratory data, including complete blood count [differential white blood cell (WBC) and platelet counts], glucose levels, and creatinine levels, were obtained on arrival at the emergency department (ED). Serum uric acid and fasting lipid profile, including cholesterol, low-density lipoprotein (LDL) cholesterol, and triglyceride levels, were obtained within 24 h of admission. Neutrophil-to-lymphocyte ratio (NLR) was calculated as the neutrophil count divided by the lymphocyte count. Data on eight risk factors, namely hypertension, diabetes mellitus, heart disease, dyslipidemia, prior stroke, current smoker, alcohol consumption (defined as the habitual consumption of alcoholic beverages at least once a week in the past year), and cancer history, were collected. We also recorded the structural time interval related to acute stroke diagnosis and treatment, namely the onset-to-ED time.

### 2.2. Statement of Ethics

This study was conducted in accordance with the Declaration of Helsinki. Ethical approval for this study was provided by the Institutional Review Board of Taipei Tzu Chi Hospital, New Taipei City (approval no. 12-X-099). The requirement for informed written consent was waived because this study involved retrospective data analysis. All the collected and analyzed data were derived from clinical records without any intervention or influence on clinical treatment. To fully protect patients’ privacy and rights, only clinical observation data have been published; personal information was not disclosed to any third party without patient consent.

### 2.3. Stroke Severity, Classification, and Clinical Features

Initial stroke severity was assessed according to the NIHSS score. According to the TOAST classification, we classified the ischemic stroke etiology into SAO, LAA, cardioembolism, other determined etiology, and undetermined etiology. Urinary tract infection, pneumonia, gastrointestinal bleeding, and seizure were recorded as stroke in-hospital complications. Neurological deterioration (ND) was defined as an increase of ≥2 points in the NIHSS score within 72 h of hospitalization, including that caused by symptomatic hemorrhagic transformation. Functional outcomes were evaluated using the NIHSS and modified Rankin Scale (mRS) scores at discharge. An mRS score of >2 was considered to indicate an unfavorable outcome.

### 2.4. Seasonal Variation and Weekday Identification

Stroke occurrence was stratified by seasons and weekdays or holidays. In this study, four seasons—spring (March–May), summer (June–August), fall (September–November), and winter (December–February)—were considered. Moreover, we defined weekdays as Mondays through Fridays, and holidays were defined as weekends (including Saturdays and Sundays) and Taiwan’s public holidays, including New Year’s Day, Lunar New Year (4 days), the Dragon Boat Festival, the Mid-Autumn Festival, Qingming Festival, Children’s day, and Taiwan’s National Day.

### 2.5. Statistical Analyses

Either the chi-square test or Fisher’s exact test was used for our comparisons of categorical variables. Continuous variables, which are presented as median (first to third quartile), were compared between groups by using the Mann–Whitney U test or Kruskal–Wallis test. Correlations between continuous variables were analyzed using Spearman’s rank coefficient correlation or least squares regression tests. We also used least squares regression to analyze the yearly secular trends of each variable. The continuous variables of age, initial NIHSS score, and laboratory data were converted into dichotomous variables by using the optimal cutoff value, determined according to the Youden index with a receiver operating characteristic (ROC) curve plotted for unfavorable outcomes and death prediction.

The variables were subsequently included in a multiple logistic regression model to identify the significant factors associated with a fatal outcome. The predictive performance of the significant variables was analyzed using C-statistics for fatal outcomes in all patients with AIS as well as those with CE stroke. A *p* value of <0.05 was considered to indicate significance. We developed nomograms for predicting the death of all patients with AIS and patients with CE by using the most significant predictors from C-statistics. All statistical analyses were performed using SPSS (version 24; IBM, Armonk, NY, USA). Scatter plot diagrams were constructed, and the ROC curves were compared using MedCalc (version 18; MedCalc Software, Mariakerke, Belgium). Nomograms were developed using STATA version 18 (StataCorp, College Station, TX, USA) and validated and calibrated using Orange version 3.36 [13].

## 3. Results

Over the 12-year study period, 4040 patients with AIS (43% female) were enrolled. Of these patients, 729 (18%) had a CE stroke. The distribution of disease characteristics regarding the 4040 patients is summarized in Table 1. The median age of onset was 71.9 years. Compared with male patients, female patients were older; they also had higher systolic blood pressure values, heart rates, platelet counts, cholesterol and LDL cholesterol levels, heart disease prevalence, cancer history prevalence, in-hospital complication rates, and CE stroke rates. Women also had longer hospital LOS values, higher initial and discharge NIHSS scores, higher mRS scores, and higher rates of unfavorable outcomes (mRS > 2). The mortality rate did not significantly differ between men and women. The median age of patients with in-hospital complications was much higher than that of patients without in-hospital complications (79.6 vs. 70.6 years; *p* < 0.001). A clear onset time of stroke was available for 1866 patients. A significant and negative correlation was observed between the onset-to-ED time and the initial NIHSS and discharge mRS scores after applying Spearman’s rank coefficient correlation [rho (95% confidence interval) = −0.292 (−0.333 to −0.250) and −0.097 (−0.141 to −0.052), respectively] and linear regression (r^2^ = 0.047 and 0.011, respectively; all *p* < 0.001).

We compared the clinical features and outcomes of 4040 patients stratified by TOAST subtypes (Table 2). Patients with CE were the oldest; had the shortest onset-to-ED times, highest heart rates, highest uric acid levels, highest heart disease rates, highest prior stroke rates, highest in-hospital complication rates, and longest hospital LOS; and received the most prescriptions for intravenous thrombolysis (IVT) and endovascular thrombectomy (EVT) treatments. By contrast, they had the lowest platelet counts, cholesterol levels, LDL cholesterol levels, and triglyceride levels, as well as the lowest dyslipidemia, current smoking, and diabetes mellitus rates. Patients with CE stroke also had the highest initial NIHSS and discharge mRS scores, as well as the highest unfavorable outcome and mortality rates. ND incidence was higher in the patients with LAA and CE stroke than in the other patients.

To analyze the influence of age, we divided patients into four age groups: <65, 65–74.9, 75–84.9, and ≥85 years. Among patients aged < 65 years, CE stroke prevalence was only 8.5%; however, it increased to 30.9% in patients aged ≥ 85 years (Figure 1). Similarly, LAA stroke prevalence increased gradually from 30.7% in patients aged < 65 years to 32.9% in those aged 75–84.9 years but remained at 32.8% in those aged ≥ 85 years. The mean numbers of risk factors, computed by the summation of the eight aforementioned risk factors, were 2.1, 2.3, 2.2, and 2.1 (range: 0–8) in patients aged > 65, 65–74.9, 75–84.9, and ≥85 years, respectively. In our least squares regression analysis, the curve of age versus number of risk factors exhibited an inverted and U-shaped pattern (Figure 2); few concomitant risk factors were observed in both younger and older patients.

In total, 261 (6.5%) patients received IVT treatment. Among them, 35.2% and 35.6% had LAA and CE stroke, respectively. These patients had similar median initial NIHSS (=15) and discharge mRS scores (=4). Death occurred in 13% of patients with LAA stroke, but only 6.5% of patients with CE stroke died; however, this result was not significant because of the small sample size.

The number of patients with AIS, regardless of the subtype, was the highest during winter (Table 3). No difference was noted in most clinical features, including stroke severity and clinical outcomes, among the four seasons. The onset-to-ED time tended to be longer in winter, but not significantly so. The median systolic blood pressure and heart rate were also the highest in winter. According to a report from the Central Weather Administration, the average temperature in 1991–2020 in Taipei (where the index hospital was located) was the lowest (16.6 °C) in winter (January) and the highest (30.1 °C) in summer (July) [14]. The average humidity was the lowest (70.2%) in July and the highest (77.2%) in January. The average wind speed was the lowest (1.9 m/s) in June and the highest (3.1 m/s) in November. The humidity exhibited a negative correlation with the temperature (rho = −0.874, *p* < 0.001). Figure 3A compares monthly temperature and humidity variations with the frequencies and percentages of patients with CE stroke. Compared with that in other months, both the frequencies and percentages of patients with CE stroke were higher in January, February, and December. Figure 3B presents the average number of all patients with AIS, specifically CE stroke per season. CE stroke prevalence significantly increased from 14.9% in summer to 23.0% in winter. In our Spearman’s rank coefficient correlation analysis, no correlation was observed between monthly variations in temperature and the number of all patients with AIS (Figure 4A). By contrast, a significant negative correlation was observed between monthly variations in temperature and the number of patients with CE stroke (Figure 4B). Similarly, a significant positive correlation was observed between monthly variations in humidity and the number of patients with CE stroke (rho = 0.754, *p* = 0.005). No differences were observed in most clinical features of patients with AIS between weekdays and holidays in terms of the stroke onset day (Table 4). The initial NIHSS and discharge mRS scores were significantly lower in patients with stroke onset on holidays (*p* = 0.045 and 0.031, respectively). Moreover, the rate of unfavorable outcomes tended to be lower in patients with stroke onset on holidays (*p* = 0.059).

ND occurred in 11% of patients with AIS; however, ND prevalence was high in patients with LAA (15%) and CE (14%) stroke (Table 1 and Table 2). No differences were noted in most risk factors between the patients with and without ND. However, compared with those without ND, patients with ND demonstrated higher heart disease (34% vs. 28%, *p* = 0.020) and AF (24% vs. 18%; *p* = 0.002) rates, higher median initial NIHSS scores (9 vs. 4; *p* < 0.001), higher median systolic blood pressure (171 vs. 160 mmHg; *p* < 0.001), and higher glucose levels (147 vs. 136 mg/dL; *p* < 0.001). Moreover, 13% of all patients with AIS had in-hospital complications, with the highest rates being found in patients with CE stroke (23%; Table 1 and Table 2). Compared with those without in-hospital complications, the patients with in-hospital complications were older (79.6 vs. 70.6 years; *p* < 0.001); had higher rates of hypertension (75% vs. 70%; *p* = 0.028), diabetes mellitus (44% vs. 35%; *p* < 0.001), and heart disease (45% vs. 27%; *p* < 0.001); higher median initial NIHSS scores (16 vs. 4; *p* < 0.001); higher median NLRs (3.9 vs. 2.9; *p* < 0.001); and higher glucose levels (146 vs. 136 mg/dL; *p* < 0.001).

We used least squares regression to analyze the yearly secular trend from 2011 to 2022 in each variable, including patient numbers, age, prevalence of CE stroke, neurological deterioration, in-hospital complications, unfavorable outcomes, and death. Given that the study period was only 12 years, most variables did not show a yearly secular trend. Only the prevalence of cardioembolism exhibited an increasing trend year by year, and the rate of in-hospital complications exhibited a decreasing yearly secular trend (Figure 5). In 2020, the height of the COVID-19 pandemic period, the number of patients admitted to hospital (*n* = 266) was well below the annual average (*n* = 337), and there was a higher rate of death (8.6%) compared to the annual average (4.7%).

Appendix A presents the results of our univariate analyses of the clinical features and unfavorable outcomes among all 4040 patients with AIS and 729 patients with CE stroke. In AIS, NLR is a more significant inflammatory marker than the WBC count; therefore, we excluded the WBC count from the outcome analysis [15]. Both the aforementioned groups of patients with unfavorable outcomes were more likely to be older in age; female; have high initial NIHSS scores, NLRs, and glucose levels; and have diabetes mellitus, a prior stroke, in-hospital complications, and ND; by contrast, they demonstrated low hemoglobin, triglyceride, and uric acid levels, as well as low current smoking and alcohol consumption rates. Among all 4040 patients with AIS, those with unfavorable outcomes had a higher heart rate, a higher heart disease rate and cancer history prevalence, a lower platelet count, lower LDL cholesterol levels, and a lower dyslipidemia rate. We converted significant continuous variables into dichotomous variables by using the optimal cutoff value, determined according to the Youden index for comparing all patients with AIS and patients with CE stroke. Table 5 presents the results of our multivariable logistic regression analysis of factors influencing unfavorable outcomes in both groups of patients. In all patients with AIS, the significant predictors of unfavorable outcomes were age > 72 years, heart rate > 86 beats/minute (BPM), initial NIHSS score > 5, NLR > 3.5, glucose level > 112 mg/dL, female sex, diabetes mellitus and prior stroke, in-hospital complications, and ND. The C-statistic of these 10 significant predictors of unfavorable outcomes was 0.865 (0.854–0.876; Table 6). Furthermore, our multiple logistic regression analysis revealed that the significant predictors of unfavorable outcomes in patients with CE stroke were age > 73 years, initial NIHSS score > 6, hemoglobin level < 12.8 g/dL, NLR > 5, in-hospital complications, and ND (Table 5). The C-statistic of these six significant predictors was 0.882 (0.856–0.905; Table 6).

Appendix A presents the results of our univariate analyses of the clinical features and death among all patients with AIS and patients with CE stroke. Both patient groups had identical significant predictors of death: older age; higher heart rates, initial NIHSS scores, NLRs, and glucose levels; cancer history; in-hospital complications; and ND. Among all patients with AIS, those with fatal outcomes had higher creatinine levels and heart disease rates; lower hemoglobin, LDL cholesterol, and triglyceride levels; and lower dyslipidemia and prior stroke rates. Among all patients with CE stroke, those with fatal outcomes had lower alcohol consumption rates. After converting continuous variables to dichotomous variables by using the optimal cutoff value, we performed a multivariable logistic regression analysis of the factors influencing death in both patient groups (Table 7). The significant predictors of death in all patients with AIS were as follows: initial NIHSS score > 10, NLR > 6, glucose level > 129 mg/dL, creatinine level > 1.11 mg/dL, LDL cholesterol level < 87 mg/dL, triglyceride level < 91 mg/dL, heart disease and cancer history, in-hospital complications, and ND. The C-statistic of these 10 significant predictors of death was 0.924 (0.915–0.932; Figure 4A); the 5 most dominant predictors were initial NIHSS score > 10, ND, NLR > 6.0, in-hospital complications, and creatinine > 1.11 mg/dL, with a predictive performance of 0.916 (Table 8). The significant predictors of death in patients with CE stroke were as follows: age > 86 years, heart rate > 79 BPM, initial NIHSS score > 16, NLR > 6.4, glucose level > 159 mg/dL, cancer history, in-hospital complications, and ND (Table 7). The C-statistic of these six significant predictors was 0.861 (0.834–0.886; Figure 4B). Of these, initial NIHSS score > 16, ND, and NLR > 6.4 were the three most dominant predictors, with a predictive performance of 0.835 (Table 8).

Based on the results of our multivariable analyses and the C-statistics, we chose the most dominant significant factors to establish nomograms for predicting death in all patients with AIS and patients with CE stroke (Figure 6). The areas under the ROC curve in our internal validation, which included the 10-fold cross-validation of the nomograms for all patients with AIS and patients with CE stroke, were 0.924 and 0.860, respectively. Further calibration plots indicated that the prediction probabilities for the nomograms referring to all patients with AIS and patients with CE stroke were 0.943 and 0.872, respectively, and the classification accuracy values were 0.955 to 0.900, respectively.

## 4. Discussion

In this study, patients with CE stroke accounted for 18.1% of all included patients with AIS. CE stroke prevalence was negatively correlated with the temperature and season, increasing gradually from its lowest point (14.9%) in summer to its highest (23%) in winter. Moreover, CE stroke prevalence was higher in female and older patients, with the highest prevalence (30.1%) being identified in patients aged ≥ 85 years. Patients with CE stroke had the shortest onset-to-ED time and the highest initial stroke severity; received the most IVT and EVT treatments; and had the most in-hospital complications, the longest hospital LOS, and the highest rates of unfavorable outcomes and mortality. The five most dominant significant predictors of unfavorable outcomes in all patients with AIS were initial NIHSS score > 5, ND, in-hospital complications, age > 72 years, and stroke history. By contrast, in patients with CE stroke, the five most significant predictors of unfavorable outcomes were initial NIHSS score > 6, ND, in-hospital complications, NLR > 5, and age > 73 years. The three most significant predictors of death in all patients with AIS were initial NIHS score > 10, ND, and NLR > 6; by contrast, in patients with CE stroke, the three most significant predictors of death were initial NIHSS score > 16, ND, and NLR > 6.4.

AF is the most common cause of CE stroke; this may be because the potential embolic source of CE stroke is the heart, and impaired cardiac function leads to reduced cerebral hemodynamics [16]. AF prevalence increases with age; approximately 82% and 37% of patients with AF are aged ≥ 65 and ≥80 years, respectively [17,18]. CE stroke prevalence also increases with age. We noted a similar result in the current study. In theory, the number of risk factors for vascular disease increases with age; simultaneously, LAA stroke prevalence increases, possibly due to atherosclerosis development and progression. Consequently, in the present study, LAA stroke prevalence increased gradually by 2.2%, changing from 30.7% in patients aged < 65 years to 32.9% in patients aged 75–84.9 years; however, it did not increase further in patients aged ≥ 85 years. By contrast, CE stroke prevalence increased considerably by 22.4%, changing from 8.5% in patients aged < 65 years to 30.9% in patients aged ≥ 85 years. The inverted U-shaped relationship between the number of risk factors and age indicated that older patients had relatively few risk factors; this partially explains the increased stroke risk from CE stroke in older patients. In living organisms, aging involves a persistent, systematic process of degeneration, which leads to the attenuation of most biochemical and physiological functions. Age, hypertension, congestive heart failure, diabetes mellitus, coronary artery disease, and valvular disease are independent risk factors for AF [19]. In this study, among all stroke patients aged ≥ 85 years, 63.7% had LAA or CE stroke. Therefore, the rates of unfavorable outcomes (81%) and death (10%) were the highest among patients with LAA or CE stroke.

Seasonal variations in stroke occurrence depend on altitude, climate, and distance from the equator. Bücke et al. reported that the higher the overall temperature, the lower the stroke incidence; as the temperature decreases, embolic stroke prevalence increases [20]. In a nationwide study, Liao et al. found seasonal variations in ischemic stroke incidence among AF patients, identifying an increased stroke risk on days with an average temperature of <20 °C [21]. Our study also showed that CE stroke prevalence was the highest in winter; it was 8% higher than that in summer. The median systolic blood pressure and heart rate were also the highest in winter. No seasonal pattern was found for other stroke subtypes. These seasonal variations may be due to an increase in AF incidence through enhanced cold-induced sympathetic function or hypertension due to low temperatures [22]. Other potential factors include cold weather-related increases in the plasma fibrinogen level, platelet count, and blood viscosity, as well as in the factor VII clotting activity [23]. Sensi et al. noted that 14.7% lower physical activity during winter can increase AF risk [9]. Stroke incidence has been reported to be lower on weekends, but patients admitted on weekends had higher stroke mortality in [24]. Although we did not analyze the circaseptan variability of stroke incidence, our results did not reveal differences in clinical features and stroke subtypes between weekdays and holidays. The reasons for the lower initial NIHSS and discharge mRS scores on holidays compared to on weekdays were unclear. Weekly variability is related to behavior or short-term lifestyle changes occurring during certain periods of the week. Extremely low or high humidity has been reported to be associated with increased hospitalizations due to stroke; however, the influence of humidity may not be as strong as that of temperature [25].

The significant predictors of unfavorable outcomes that were common between all patients with AIS and patients with CE stroke were older age, female sex, high initial NIHSS score and NLR, ND, and in-hospital complications. By contrast, the significant predictors of death that were common between the two patient groups were high initial NIHSS score and NLR, ND, and cancer history. The rate of in-hospital complications was a significant predictor of death in all patients with AIS, but not in patients with CE stroke.

Initial stroke severity was the most significant predictor of both unfavorable outcomes and death. The optimal initial NIHSS score cutoffs in patients with AIS and CE stroke were >5 and >6 for unfavorable outcomes and >10 and >16 for death, respectively. Patients with CE stroke had the highest initial NIHSS score. Therefore, their optimal initial NIHSS cutoff for death was much higher than that in patients with AIS. Given that CE stroke usually presents as a neurologic deficit with abrupt onset which reaches maximal intensity within a few minutes, patients with CE stroke are more likely to be recognized and admitted to the ED earlier. A higher initial stroke severity and a shorter onset-to-ED time increase the likelihood of patients with CE stroke receiving IVT or EVT treatment. Although the patients with CE stroke had the highest initial NIHSS score, they also achieved the greatest median NIHSS score improvements at discharge—from 10 to 6. The post-IVT mortality in patients with LAA stroke was twice that in patients with CE stroke; in other words, IVT led to more favorable outcomes in patients with CE stroke than in patients with LAA stroke. The post-IVT clinical outcomes in patients with CE stroke that have been reported thus far have been controversial. Compared with red blood cell-rich atherosclerosis thrombi, CE thrombi have higher fibrin- and platelet-rich components, which are more difficult to dissolve through IVT or to retrieve through EVT [26]. Some studies have emphasized that post-IVT outcomes are worse in patients with CE stroke than in patients with either non-CE or non-LAA stroke [27]. Several studies have reported improved outcomes in patients with CE stroke receiving IVT treatment [28,29]. We noted that the mortality rate in patients with CE stroke was half of that in patients with LAA stroke; relevant studies with larger patient samples are warranted to confirm this finding.

ND occurrence has been reported in up to 40% of all patients with AIS [30]. The definition and etiology of ND vary among studies. In the present study, we defined ND as an increase of ≥2 points in the NIHSS score within 72 h of AIS occurrence, which is caused by ischemic area extension, recurrent stroke, or symptomatic intracerebral hemorrhagic transformation. This definition is similar to that of nonreversible ND reported by Siegler et al. [31]. A large infarct area with brain edema and subsequent herniation during AIS was not considered an indicator of early ND in the present study. In this study, ND prevalence was high among patients with LAA or CE stroke, and ND was a strong predictor of both unfavorable outcomes and death in all patients with AIS and patients with CE stroke. Although high systolic blood pressure was associated with ND in our study, Toyoda et al. reported that systolic blood pressure values 12–36 h after admission, but not those at admission or 6 h after admission, are predictive of ND within the initial 3 weeks of an ischemic stroke, particularly in patients with CE stroke [32]. Chung et al. also reported that blood pressure variability is independently and linearly associated with ND development [33]. ND risk increases as the variability in each blood pressure parameter increases. Acute blood pressure reduction did not improve stroke outcomes. Frequent blood pressure monitoring after AIS occurrence may be more appropriate for the detection of blood pressure variability. Moreover, high glucose levels are associated with ND occurrence [34]. Hyperglycemia may increase brain lactate levels and thus aggravate infarction in the hypoperfused brain tissue. The prothrombotic effects of hyperglycemia may also lead to infarct area extension.

In-hospital complications were a major predictor of unfavorable outcomes in our patients with AIS and CE stroke; they were also a major predictor of death in all patients with AIS. Pneumonia and urinary tract infection were the two most common causes of in-hospital complications. As age increases, stroke complication risk increases [35]. In-hospital complications most commonly occurred in patients with CE stroke (23%) and patients aged ≥ 85 years (25%). Similar to ND, hyperglycemia was associated with the development of in-hospital complications. An elevated NLR may reflect the activation of both innate and adaptive immune responses during acute stroke through sympathetic activation and hypothalamic–pituitary–adrenal axis activation. This stress-induced immune alteration may lead to stroke-associated infections [15]. Inflammatory marker levels increase linearly with the initial NIHSS and discharge mRS scores [15]. In the current study, CE and LAA strokes with high stroke severity (as indicated by high NIHSS scores) accounted for 50% of all AIS cases. CE stroke prevalence was higher in older patients. Older patients with greater disease severity and multiple comorbidities typically have numerous complications. Stroke-induced immunosuppression also increases post-stroke infection susceptibility. Post-stroke infections further increase the inflammatory response, increasing disease severity. This vicious cycle leads to poor clinical outcomes in the affected patients.

Notably, in the present study, cancer history was a predictor of death but not of unfavorable outcomes. Sonbel et al. reported that the risk of stroke mortality in cancer patients is significantly higher than that in the general population, with the risk being higher in patients with colorectal or lung cancer [36]. The possible mechanisms underlying stroke development in patients with cancer include hypercoagulability, nonbacterial thrombotic endocarditis, direct blood vessel compression, and radiotherapy. Lun et al. reported that stroke risk increases earlier in the cancer course, particularly within the first year [37]. However, the highest mortality rate was observed 1–5 years after stroke diagnosis [38].

Prediction models with reliable nomograms provide better prediction performance for patient outcomes. In the present study, we developed nomograms to better identify patients at risk of death with good discriminative ability and calibration. We could predict the mortality of patients by using the nomograms using the corresponding categorial and continuous variables. For instance, an 82-year-old woman with a history of AF and cancer an initial NIHSS score of 25, a heart rate of 100 BPM, a serum creatinine level of 2.1 mg/dL, and an NLR value of 6.3 experienced ND within two days of hospitalization and, subsequently, pneumonia. This gave her a total score of 13.9 points based on the nomogram for all ischemic stroke patients (Figure 6A), corresponding to a probability of death of approximately 0.83 (83%); she also had a total score of 24.3 based on the nomogram for patients with CE stroke (Figure 6B), corresponding to a probability of death of approximately 0.79 (79%). The predicted probabilities of death from CE stroke are similar in both nomograms.

This study has several limitations. First, hemorrhagic transformation was observed in 0.3% of all patients with AIS, with the highest rate being observed in those with CE stroke (6%). Some early onset symptomatic hemorrhagic transformations were included in ND. However, we did not include hemorrhagic transformation in our analyses because not all patients received follow-up brain imaging, and asymptomatic hemorrhagic transformation could not be identified. Thus, predicting the actual frequency of hemorrhagic transformation is difficult. Second, symptom onset times were available for only 46% of patients, which might have led to bias in our statistical analyses. Patients with available onset time were admitted to the ED earlier, and a code stroke was initiated for patients arriving at the ED within the therapeutic windows for IVT or EVT. The third quartile of the onset-to-ED time was 633 min, equivalent to approximately 10.5 h. Based on experience, a clear symptom onset time is generally difficult to confirm, particularly in patients with insidious symptom onset and those who arrive at the ED > 10 h after symptom onset. Additionally, the patient’s response time between the onset of symptoms has been reported to be related to behavioral, cognitive, and contextual factors [39]. Third, the influence of temperature on CE stroke incidence varies in different areas. We did not compare the seasonal variations in CE stroke prevalence among different areas in Taiwan; for instance, in southern Taiwan (nearly 350 km away from the study location), the monthly change in mean temperature is narrow, ranging from 19.7 to 29.4 °C. Fourth, the rates of patients receiving IVT (6.5%) and EVT (2.5%) treatments in this study were relatively low. Through serial interventions for reducing in-hospital treatment delays and the continuous updating of applicable guidelines [40], the IVT rates at the index hospital have increased from 3.2% in 2011 to 14% in 2023. A formal EVT treatment program was launched at the index hospital in May 2017. Following the recommendations from the reimbursement criteria of the Bureau of National Health Insurance, patients who presented to the ED within 8 h of symptom onset with an NIHSS score of 8–30 were eligible for EVT treatment. The current EVT rate is 6.5%. Since November 2023, the golden period for receiving EVT treatment has been extended to 24 h for patients with NIHSS scores of 6–30. We applied the newly established DARE-PACE assessment at the ED to shorten delays in in-hospital time in order to treat more patients during the hyperacute stage of ischemic stroke [41]. Finally, our data included limited information on discharge outcomes. Therefore, we could not determine an intermediate outcome at 3 months or conduct long-term follow-up.

## 5. Conclusions

CE stroke most commonly occurs during winter and in individuals aged ≥ 85 years. The three most dominant factors influencing death, noted in not only patients with AIS but also those with CE stroke, are a high initial NIHSS score, the presence of ND, and a high NLR. However, compared with patients with AIS, patients with CE stroke exhibit higher initial stroke severity, higher in-hospital complication rates, more unfavorable outcomes, and higher mortality. To detect AF earlier and therefore prevent CE stroke, comprehensively monitoring patients’ heart rhythms, particularly in older patients and during winter, is highly warranted.

## Figures and Tables

**Figure 1 biomedicines-12-00223-f001:**
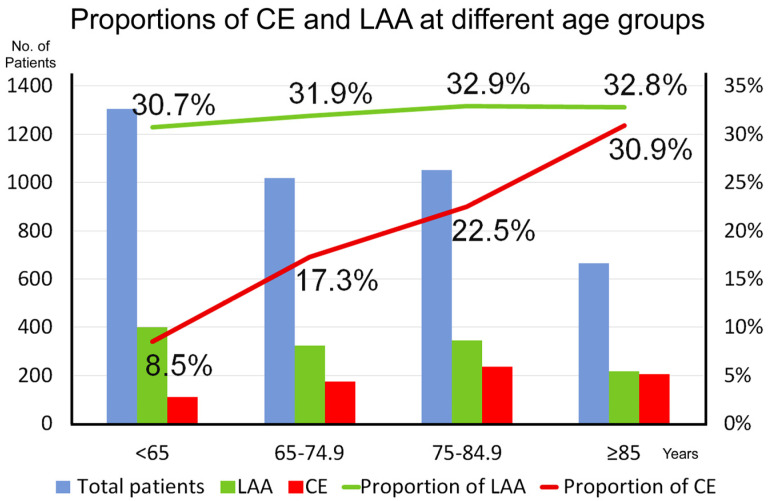
The prevalence of cardioembolic (CE) stroke increased from 8.5% among patients aged < 65 years to 30.9% among patients aged ≥ 85 years. No prominent increment in the prevalence of large-artery atherosclerosis (LAA) was observed among the different age groups.

**Figure 2 biomedicines-12-00223-f002:**
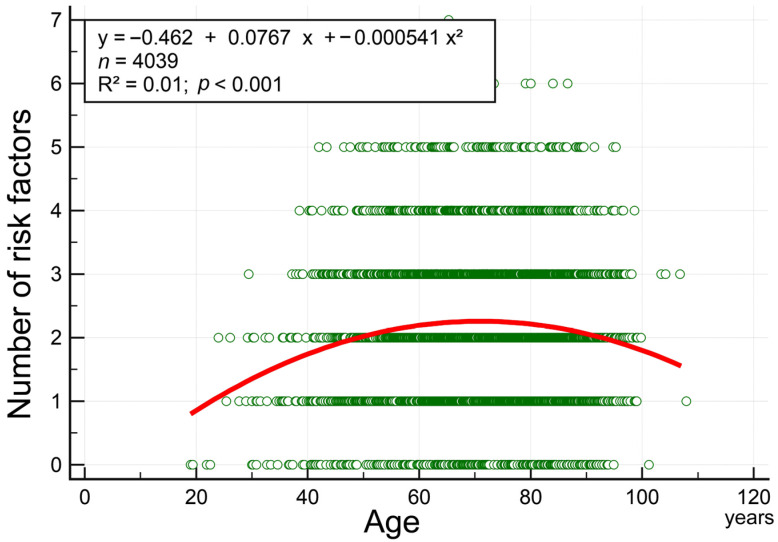
An inverted U-shaped relationship exists between the number of risk factors and age.

**Figure 3 biomedicines-12-00223-f003:**
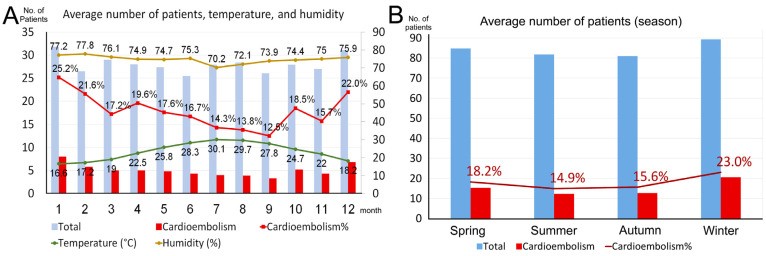
(**A**) The monthly distribution of the average number of all patients with acute ischemic stroke and patients with cardioembolic stroke. The temperature and humidity are displayed with the number and percentages of patients. (**B**) The seasonal distribution of the number of all patients with acute ischemic stroke and patients with cardioembolic stroke displayed with the number and percentages of patients.

**Figure 4 biomedicines-12-00223-f004:**
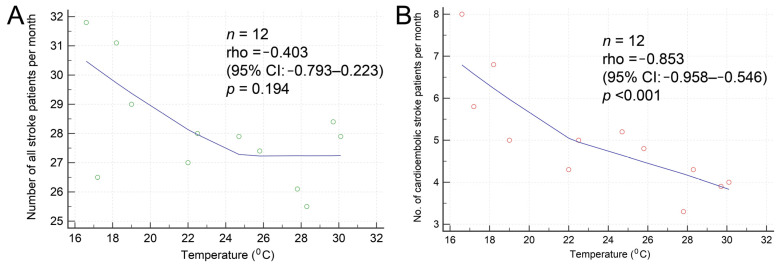
(**A**) No significant correlation was found between the temperature and the number of all stroke patients per month. (**B**) A significant negative correlation exists between the temperature and the number of patients with cardioembolic stroke per month.

**Figure 5 biomedicines-12-00223-f005:**
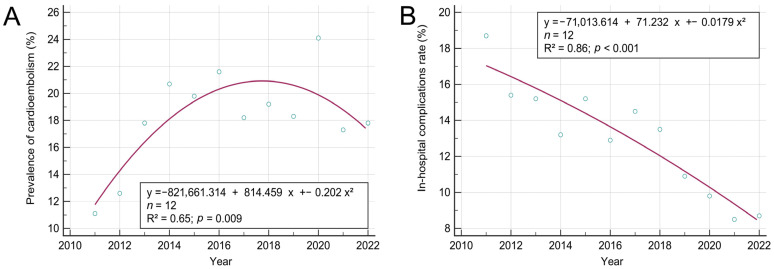
(**A**) The prevalence of cardioembolism exhibits an increasing secular trend year by year. (**B**) The rate of in-hospital complications exhibits a decreasing yearly secular trend.

**Figure 6 biomedicines-12-00223-f006:**
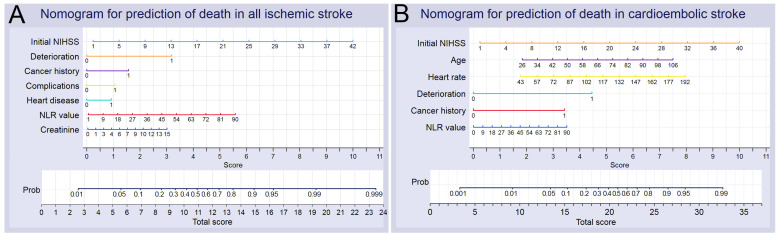
Nomograms for predicting death in all patients with acute ischemic stroke (**A**) and in patients with cardioembolic stroke (**B**). A vertical line was drawn from the value of each variable down to the “Score” line to match the scores, and the matched scores were summed to obtain a total score. Then, a vertical line was drawn from the “Total Score” up to the “Prob” line to match the appropriate probability of death. Prob, probability; NIHSS, National Institutes of Health Stroke Scale; NLR, neutrophil-to-lymphocyte ratio.

**Table 1 biomedicines-12-00223-t001:** Baseline data and clinical features of 4040 patients with acute ischemic stroke.

Characteristics	All Patients (*n* = 4040)	Gender		*p*-Value
		Men (*n* = 2299)	Women (*n* = 1741)	
Age (years)	71.9 (62.2–81.5)	68.5 (59.7–78.8)	76.4 (66.0–84.4)	<0.001
Onset-to-ED time (min; *n* = 1866)	278 (70–633)	277 (85–642)	281 (197–625)	0.759
Systolic blood pressure (mmHg)	161 (142–185)	160 (142–182)	163 (142–188)	0.009
Heart rate (beats/minute)	80 (69–92)	79 (68–91)	80 (70–92)	0.008
Hemoglobulin (g/dL)	13.7 (12.3–15.0)	14.4 (13.1–15.5)	12.9 (11.6–14.0)	<0.001
Platelet (×10^9^/L)	208 (169–254)	203 (164–247)	216 (179–265)	<0.001
White blood cells (×10^3^/mL)	7.60 (6.17–9.50)	7.70 (6.21–9.56)	7.50 (6.10–9.35)	0.036
Neutrophil-to-lymphocyte ratio	2.90 (2.0–4.8)	3.0 (2.1–4.8)	2.9 (1.9–4.8)	0.340
Glucose (mg/dL)	138 (112–189)	138 (113–187)	138 (112–190)	0.952
Creatinine (mg/dL)	1.05 (0.87–1.30)	1.10 (0.96–1.40)	0.90 (0.71–1.20)	<0.001
Cholesterol (mg/dL)	165 (140–193)	162 (137–189)	170 (143–200)	<0.001
LDL cholesterol (mg/dL)	104 (82–128)	102 (81–128)	105 (84–129)	0.095
Triglyceride (mg/dL)	103 (74–149)	103 (74–153)	104 (74–144)	0.340
Uric acid (mg/dL)	5.1 (4.1–6.2)	5.4 (4.4–6.5)	4.7 (3.8–5.9)	<0.001
Hypertension	2863 (71)	1608 (70)	1255 (72)	0.138
Diabetes mellitus	1467 (36)	818 (36)	649 (37)	0.267
Heart disease	1168 (29)	618 (27)	510 (32)	0.001
Dyslipidemia	878 (22)	492 (21)	386 (22)	0.557
Prior stroke	962 (24)	576 (25)	386 (22)	0.033
Current smoker	853 (21)	793 (35)	60 (3)	<0.001
Alcohol consumption	257 (7)	252 (11)	5 (0.3)	<0.001
History of cancer	275 (7)	127 (6)	148 (9)	<0.001
In-hospital complications	509 (13)	235 (10)	274 (16)	<0.001
Neurological deterioration	425 (11)	229 (10)	196 (11)	0.183
TOAST classification				<0.001
Small-artery occlusion	1799 (44.5)	1088 (47.3)	711 (40.9)	
Large-artery atherosclerosis	1288 (31.9)	726 (31.6)	562 (32.3)	
Cardioembolism	729 (18.1)	363 (15.8)	366 (21.0)	
Other determined etiology	81 (2.0)	30 (1.3)	51 (2.9)	
Undetermined etiology	143 (3.5)	92 (4.0)	51 (2.9)	
Intravenous thrombolytic therapy	261 (6.5)	165 (7.2)	96 (5.5)	0.033
Endovascular thrombectomy therapy	99 (2.5)	59 (2.6)	40 (2.3)	0.609
Length of stay (days)	10 (5–21)	9 (5–19)	12 (6–22)	<0.001
Initial NIHSS score	5 (2–11)	4 (2–9)	6 (3–13)	<0.001
Discharge modified Rankin Scale score	3 (1–4)	2 (1–4)	4 (2–4)	<0.001
Discharge modified Rankin Scale score > 2	2262 (56)	1124 (49)	1138 (65)	<0.001
Death at discharge	191 (4.7)	97 (4.2)	94 (5.4)	0.081

Data are expressed as median (25–75 percentile) or *n* (%). ED, emergency department; LDL, low-density lipoprotein; NIHSS, National Institutes of Health Stroke Scale; TOAST, Trial of ORG 10172 in Acute Stroke Treatment.

**Table 2 biomedicines-12-00223-t002:** Clinical features according to the TOAST classification in 4040 patients with acute ischemic stroke.

Characteristics	SAO (*n* = 1798)	LAA (*n* = 1288)	CE (*n* = 729)	OD (*n* = 81)	UD (*n* = 143)	*p*-Value
Age (years)	69.1 (60.5–79.0)	72.6 (62.2–81.8)	78.5 (69.2–85.7)	60.1 (50.1–78.9)	69.0 (60.5–80.4)	<0.001
Female sex	711 (40)	562 (44)	366 (50)	51 (63)	51 (36)	<0.001
Onset-to-ED (min; *n* = 1866)	388 (149–793)	271 (86–621)	166 (50–404)	215 (97–374)	236 (49–581)	<0.001
Systolic blood pressure (mmHg)	163 (144–186)	162 (142–186)	157 (137–178)	144 (132–171)	159 (140–186)	<0.001
Heart rate (Beats/minute)	78 (68–89)	80 (69–91)	87(76–98)	85(72–98)	82 (69–93)	<0.001
Hemoglobulin (g/dL)	13.8 (12.6–15.0)	13.7 (12.2–15.0)	13.5 (12.1–15.0)	12.9 (108–14.4)	13.8 (12.1–15.2)	<0.001
Platelet (×10^9^/L)	211 (176–256)	212 (171–260)	190 (155–231)	240 (180–240)	204 (162–252)	<0.001
White blood cells (×10^3^/mL)	7.32 (6.05–8.98)	7.98 (6.49–10.1)	7.45 (5.92–9.49)	8.22 (6.65–10.5)	7.97 (6.15–9.94)	<0.001
Neutrophil-to-lymphocyte ratio	2.7 (2.0–4.1)	3.3 (2.1–5.3)	3.1 (2.0–5.6)	3.8 (2.6–6.1)	3.2 (1.8–5.0)	<0.001
Glucose (mg/dL)	136 (110–196)	142 (115–199)	134 (113–169)	124 (107–173)	143 (116–175)	<0.001
Creatinine (mg/dL)	1.00 (0.82–1.30)	1.10 (0.89–1.40)	1.10 (0.90–1.40)	1.00 (0.79–1.30)	1.00 (0.80–1.40)	<0.001
Cholesterol (mg/dL)	171 (147–199)	165 (138–195)	154 (129–177)	157 (140–186)	155 (132–180)	<0.001
LDL cholesterol (mg/dL)	109 (88–133)	104 (81–130)	93 (73–114)	97 (85–120)	94 (77–121)	<0.001
Triglyceride (mg/dL)	116 (80–163)	104 (75–147)	82 (60–114)	99 (74–157)	100 (76–149)	<0.001
Uric acid (mg/dL)	5.0 (4.1–6.1)	5.1 (4.0–6.2)	5.4 (4.3–6.6)	4.9 (3.7–6.3)	5.0 (4.–6.4)	0.003
Hypertension	1290 (72)	923 (72)	514 (71)	46 (57)	90 (63)	0.014
Diabetes mellitus	672 (37)	509 (40)	221 (30)	14 (17)	51 (36)	<0.001
Heart disease	276 (15)	281 (22)	555 (78)	14 (17)	42 (30)	<0.001
Dyslipidemia	474 (26)	281 (22)	82 (11)	15 (19)	26 (18)	<0.001
Prior stroke	417 (23)	318 (25)	190 (26)	12 (15)	25 (17)	0.035
Current smoker	419 (23)	287 (22)	94 (13)	11 (14)	42 (29)	<0.001
Alcohol consumption	124 (7)	90 (7)	35 (5)	2 (2)	6 (4)	0.066
Cancer history	84 (5)	97 (8)	61 (8)	21 (26)	12 (8)	<0.001
Intravenous thrombolysis	56 (3)	92 (7)	93 (13)	4 (5)	16 (11)	<0.001
Endovascular thrombectomy	0 (0)	50 (4)	44 (7)	4 (5)	1 (1)	<0.001
In-hospital complications	100 (6)	208 (16)	169 (23)	14 (17)	18 (13)	<0.001
Neurological deterioration	106 (6)	192 (15)	103 (14)	7 (9)	17 (12)	<0.001
Length of stay (days)	8 (5–16)	13 (6–24)	14 (7–26)	12 (7–22)	8 (5–18)	<0.001
Initial NIHSS score	3 (2–6)	7 (3–15)	10 (4–19)	6 (2–12)	5 (2–14)	<0.001
Discharge NIHSS score	2 (1–4)	5 (2–13)	6 (2–18)	4 (1–11)	3 (1–10)	<0.001
Discharge mRS score	2 (1–4)	4 (2–5)	4 (2–5)	3 (2–4)	4 (1–4)	<0.001
Discharge mRS score > 2	762 (42)	875 (68)	500 (69)	47 (58)	78 (55)	<0.001
Death at discharge	4 (0.2)	93 (7)	73 (10)	8 (10)	13 (9)	<0.001

Data are expressed as median (25–75 percentile) or *n* (%). CE, cardioembolic; ED, emergency department; LAA, large-artery atherosclerosis; LDL, low-density lipoprotein; mRS, modified Rankin Scale; OD, other determined etiology; NIHSS, National Institutes of Health Stroke Scale; TOAST, Trial of ORG 10172 in Acute Stroke Treatment; SAO, small-artery occlusion; UD, undetermined etiology.

**Table 3 biomedicines-12-00223-t003:** The seasonal variation in the clinical features of 4040 patients with acute ischemic stroke.

Characteristics	Spring (*n* = 1013)	Summer (*n* = 982)	Autumn (*n* = 972)	Winter (*n* = 1073)	*p*-Value
Age (years)	72.5 (63.2–81.3)	71.1(61.6–81.3)	71.8 (62.2–81.4)	72.1 (61.3–81.8)	0.542
Onset-to-ED (min; *n* = 1866)	248 (91–653)	276 (87–663)	265 (78–545)	321 (99–651)	0.084
Systolic blood pressure (mmHg)	163 (142–186)	157 (137–180)	161 (143–183)	165 (145–189)	<0.001
Heart rate (Beats/minute)	80 (70–92)	79 (68–90)	79 (67–90)	82 (70–94)	<0.001
Intravenous thrombolytic therapy	75 (7.4)	62 (6.3)	64 (6.6)	60 (5.6)	0.409
Endovascular thrombectomy therapy	27 (2.7)	25 (2.6)	30 (3.1)	17 (16)	0.155
In-hospital complications	139 (14)	130 (13)	118 (12)	122 (11)	0.365
Neurological deterioration	107 (11)	104 (11)	102 (11)	112 (10)	0.999
TOAST classification					<0.001
Small-artery occlusion (*n* = 1799)	429 (42.3)	437 (44.5)	459 (47.2)	474 (44.0)	
Large-artery atherosclerosis (*n* = 1288)	337 (33.3)	347 (35.3)	308 (31.7)	296 (27.6)	
Cardioembolism (*n* = 729)	184 (18.2)	146 (14.9)	152 (15.7)	207 (23.0)	
Other determined etiology (*n* = 81)	23 (2.3)	13 (1.3)	17 (1.7)	28 (2.6)	
Undetermined etiology (*n* = 143)	40 (3.9)	39 (4.0)	36 (3.7)	28 (2.6)	
Initial NIHSS score	5 (3–11)	5 (2–10)	5 (2–10)	5 (2–10)	0.339
Discharge modified Rankin Scale score	3 (1–4)	3 (1–4)	3 (1–4)	3 (1–4)	0.833
Discharge modified Rankin Scale score > 2	569 (56)	547 (56)	550 (57)	596 (56)	0.965
Death	58 (5.7)	46 (4.7)	34 (3.5)	53 (4.9)	0.122

Data are expressed as median (25–75 percentile) or *n* (%). ED, emergency department; NIHSS, National Institutes of Health Stroke Scale; TOAST, Trial of ORG 10172 in Acute Stroke Treatment.

**Table 4 biomedicines-12-00223-t004:** Comparison of clinical features between weekdays and holidays in terms of stroke onset day in 4040 patients with acute ischemic stroke.

Characteristics	Weekdays (*n* = 2942)	Holidays (*n* = 1098)	*p*-Value
Age (years)	71.9 (62.4–81.4)	71.4 (61.6–81.9)	0.755
Onset-to-ED (min; *n* = 1866)	281 (94–642)	272 (78–603)	0.269
Intravenous thrombolytic therapy	196 (7)	65 (6)	0.429
Endovascular thrombectomy therapy	75 (3)	24 (2)	0.568
In-hospital complications	369 (13)	140 (13)	0.873
Neurological deterioration	312 (11)	113 (10)	0.818
TOAST classification			0.406
Small-vessel occlusion	1317 (45)	482 (44)	
Large-artery atherosclerosis	933 (32)	355 (32)	
Cardioembolism	521 (18)	208 (19)	
Other determined etiology	66 (2)	15 (1)	
Undetermined etiology	105 (4)	38 (3)	
Initial NIHSS score	5 (3–11)	4 (2–10)	0.045
Discharge modified Rankin Scale score	3 (1–4)	3 (1–4)	0.031
Discharge modified Rankin Scale score > 2	1674 (57)	588 (54)	0.059
Death	143 (5)	48 (4)	0.560

Data are expressed as median (25–75 percentile) or *n* (%). ED, emergency department; NIHSS, National Institutes of Health Stroke Scale; TOAST, Trial of ORG 10172 in Acute Stroke Treatment.

**Table 5 biomedicines-12-00223-t005:** Multivariable analysis of factors influencing unfavorable outcomes (modified Rankin Scale > 2) in all 4040 patients with acute ischemic stroke and 729 patients with cardioembolic stroke.

All 4040 Patient with Acute Ischemic Stroke			729 Patients with Cardioembolic Stroke		
Characteristics	Odds Ratio (95% CI)	*p*-Value	Characteristics	Odds Ratio (95% CI)	*p*-Value
Age > 72 years	2.594 (2.170–3.102)	<0.001	Age > 73 years	2.230 (1.431–3.474)	<0.001
Heart rate > 86 (beats/minute)	1.203 (1.009–1.435)	0.039	Initial NIHSS score > 6	9.423 (6.118–14.514)	<0.001
Initial NIHSS score > 5	10.067 (8.414–12.045)	<0.001	Hemoglobin < 12.8 g/dL	1.804 (1.109–2.934)	0.018
Hemoglobin < 13.5 g/dL	1.170 (0.977–1.401)	0.088	NLR > 5	2.356 (1.427–3.889)	<0.001
Platelet < 183 (×10^9^/L)	1.128 (0.939–1.356)	0.198	Glucose > 125 mg/dL	1.270 (0.825–1.954)	0.277
NLR > 3.5	1.562 (1.317–1.853)	<0.001	Triglyceride < 107 mg/dL	1.206 (0.763–1.906)	0.422
Glucose > 112 mg/dL	1.300 (1.079–1.566)	0.006	Female gender	1.008 (0.631–1.610)	0.974
LDL cholesterol < 88 mg/dL	0.935 (0.777–1.125)	0.478	LDL cholesterol < 88 mg/dL	0.947 (0.788–1.138)	0.563
Triglyceride < 110 mg/dL	1.121 (0.939–1.338)	0.208	Diabetes mellitus	1.413 (0.866–2.305)	0.167
Female gender	1.430 (1.184–1.728)	<0.001	Smoking	0.989 (0.504–1.940)	0.974
Diabetes mellitus	1.396 (1.165–1.672)	<0.001	Alcohol consumption	0.781 (0.292–2.087)	0.621
Heart disease	0.884 (0.729–1.071)	0.207	In-hospital complications	6.264 (2.380–16.483)	<0.001
Dyslipidemia	1.006 (0.822–1.233)	0.951	Deterioration	15.689 (4.426–55.615)	<0.001
Prior stroke	1.583 (1.304–1.922)	<0.001			
Smoking	0.817 (0.652–1.023)	0.078			
Alcohol consumption	0.832 (0.580–1.193)	0.317			
Cancer history	1.353 (0.972–1.884)	0.073			
In-hospital complications	7.156 (4.377–11.700)	<0.001			
Neurological deterioration	7.475 (5.142–10.867)	<0.001			

CI, confidence interval; NIHSS, National Institutes of Health Stroke Scale; NLR, neutrophil-to-lymphocyte ratio.

**Table 6 biomedicines-12-00223-t006:** C-statistics for unfavorable outcome (modified Rankin Scale > 2) predictions in all 4040 patients with acute ischemic stroke and 729 patients with cardioembolic stroke.

All 4040 Patient with Acute Ischemic Stroke			729 Patients with Cardioembolic Stroke		
Characteristics	C-Statistics (95% CI)	*p*-Value ^a^	Characteristics	C-Statistics (95% CI)	*p*-Value ^a^
Initial NIHSS score > 5	0.772 (0.759–0.785)		Initial NIHSS score > 6	0.788 (0.752–0.825)	
Includes neurological deterioration	0.796 (0.783–0.809)	<0.001	Includes neurological deterioration	0.816 (0.785–0.843)	<0.001
Further includes in-hospital complications	0.811 (0.799–0.823)	<0.001	Further includes in-hospital complications	0.842 (0.813–0.867)	<0.001
Further includes age > 72 years	0.849 (0.837–0.860)	<0.001	Further includes NLR > 5	0.860 (0.833–0.885)	<0.001
Further includes prior stroke	0.854 (0.842–0.864)	<0.001	Further includes age > 73 years	0.875 (0.849–0.898)	0.012
Further includes female gender	0.859 (0.847–0.869)	<0.001	Further includes hemoglobin < 12.8 g/L	0.882 (0.856–0.905)	0.044
Further includes diabetes mellitus	0.861 (0.850–0.872)	0.001			
Further includes NLR > 3.5	0.864 (0.853–0.874)	0.054			
Further includes glucose > 112 mg/dL	0.865 (0.854–0.875)	0.067			
Further includes heart rate > 86 BPM	0.865 (0.854–0.876)	0.467			

^a^ Compared with the previous one. BPM, beats per minute; CI, confidence interval; NIHSS, National Institutes of Health Stroke Scale; NLR, neutrophil-to-lymphocyte ratio.

**Table 7 biomedicines-12-00223-t007:** Multivariable analysis of death predictors in all 4040 patients with acute ischemic stroke and 729 patients with cardioembolic stroke.

All 4040 Patient with Acute Ischemic Stroke			729 Patients with Cardioembolic Stroke		
Characteristics	Odds Ratio (95% CI)	*p*-Value	Characteristics	Odds Ratio (95% CI)	*p*-Value
Age > 77 years	1.092 (0.763–1.565)	0.630	Age > 86 years	2.123 (1.167–3.862)	0.014
Heart rate > 87 beats/minute	1.277 (0.903–1.805)	0.167	Heart rate > 79 beats/minute	1.949 (1.032–3.680)	0.039
Initial NIHSS score > 10	12.810 (7.908–20.750)	<0.001	Initial NIHSS score > 16	5.820 (3.080–10.996)	<0.001
Hemoglobin < 12.2 g/dL	1.185 (0.812–1.730)	0.379	NLR > 6.4	2.359 (1.282–4.341)	0.006
NLR > 6	2.247 (1.568–3.220)	<0.001	Glucose > 159 mg/dL	1.451 (0.805–2.616)	0.215
Glucose > 129 mg/dL	1.865 (1.283–2.711)	0.001	Cancer history	4.216 (1.939–9.168)	<0.001
Creatinine > 1.11 gm/dL	1.549 (1.090–2.200)	0.015	In-hospital complications	1.591 (0.885–2.862)	0.121
LDL cholesterol < 87 mg/dL	1.633 (0.129–2.362)	0.009	Neurological deterioration	5.737 (3.104–10.604)	<0.001
Triglyceride < 91 mg/dL	1.600 (1.091–2.348)	0.016			
Heart disease	1.660 (1.172–2.353)	0.004			
Dyslipidemia	0.955 (0.561–1.625)	0.865			
Cancer history	2.238 (1.314–3.8147)	0.003			
In-hospital complications	1.790 (1.255–2.553)	0.001			
Neurological deterioration	5.967 (4.133–8.614)	<0.001			

CI, confidence interval; NIHSS, National Institutes of Health Stroke Scale; NLR, neutrophil-to-lymphocyte ratio.

**Table 8 biomedicines-12-00223-t008:** C-statistics for death predictors in all 4040 patients with acute ischemic stroke and 729 patients with cardioembolic stroke.

All 4040 Patient with Acute Ischemic Stroke			729 Patients with Cardioembolic Stroke		
Characteristics	C-Statistics (95% CI)	*p*-Value ^a^	Characteristics	C-Statistics (95% CI)	*p*-Value ^a^
Initial NIHSS score > 10	0.832 (0.820–0.844)		Initial NIHSS score > 16	0.752 (0.719–0.783)	
Includes neurological deterioration	0.877 (0.867–0.887)	<0.001	Includes neurological deterioration	0.806 (0.776–0.835)	<0.001
Further includes NLR > 6.0	0.897 (0.887–0.906)	<0.001	Further includes NLR > 6.4	0.835 (0.806–0.862)	0.007
Further includes in-hospital complications	0.909 (0.900–0.918)	<0.001	Further includes cancer history	0.851 (0.823–0.876)	0.091
Further includes creatinine > 1.11 mg/dL	0.916 (0.907–0.927)	0.003	Further includes age > 86 years	0.858 (0.831–0.883)	0.092
Further includes heart disease	0.919 (0.910–0.927)	0.266	Further includes heart rate > 79 BPM	0.861 (0.834–0.886)	1.000
Further includes cancer history	0.922 (0.913–0.931)	0.318			
Further includes triglyceride < 91 mg/dL	0.923 (0.914–0.931)	0.421			
Further includes glucose > 129 mg/dL	0.923 (0.915–0.931)	0.818			
Further includes LDL cholesterol < 87 mg/dL	0.924 (0.915–0.932)	0.729			

^a^ Compared with the previous one. BPM, beats per minute; CI, confidence interval; NIHSS, National Institutes of Health Stroke Scale; NLR, neutrophil-to-lymphocyte ratio.

## Data Availability

The data presented in this study are available from the corresponding author upon request.

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
