# Peer review of "Seasonal Variations in Stroke and a Comparison of the Predictors of Unfavorable Outcomes among Patients with Acute Ischemic Stroke and Cardioembolic Stroke"

_biomedicines, 2024, doi:10.3390/biomedicines12010223_

Round 1

Reviewer 1 Report

Comments and Suggestions for Authors

The topic of cardioembolic stroke has been studied for many years, but it remains relevant from a clinical and scientific point of view. The influence of environmental factors may play a role in the development of stroke and its outcomes. The influence of seasonal environmental factors can vary in different regions of the world and in different ethnic groups of the population, even within the same country. This explains the relevance of the authors' research topic. However, this article needs a major revision.

I recommend that the authors modify the title of the article and focus on the problem of cardioembolic stroke. The introduction needs a major revision. In this section, it is necessary to demonstrate the unresolved issues of the influence of seasonal fluctuations on the development of ischemic stroke in general and its cardioembolic subtype in particular. It is also necessary to add the purpose of this study.

Materials and methods: How was the sample size calculated? What are the inclusion and exclusion criteria? What are the criteria for the formation of observation groups? Which ethnic groups were included in this study? What are the characteristics of the seasons in Taiwan (average air temperature, average humidity, wind speed, etc.)?

Results: Add explanations of all the abbreviations that you used in Tables 1, 2. Discussion: What is the difference between the influence of the season on the incidence of cardioembolic stroke in Taiwan and other countries of Southeast Asia and other regions of the world?

It is also necessary to update links to articles published more than 10 years ago.

Comments on the Quality of English Language

A minor correction of the English language style is desirable.

Author Response

Response to Reviewer 1 Comments

Manuscript ID: Biomedicines-2755575

Title: Seasonal Variations and Comparison of Predictors of Unfavorable Outcomes Among Patients With Acute Ischemic Stroke and Those With Cardioembolic Stroke

Thanks to reviewer’s precious comments. We have checked the manuscript and have made essential revisions according to reviewer’s comments point-by-point. The revised portions in the manuscript were coded in red color.

Comments and Suggestions for Authors

The topic of cardioembolic stroke has been studied for many years, but it remains relevant from a clinical and scientific point of view. The influence of environmental factors may play a role in the development of stroke and its outcomes. The influence of seasonal environmental factors can vary in different regions of the world and in different ethnic groups of the population, even within the same country. This explains the relevance of the authors' research topic. However, this article needs a major revision.

Point 1: I recommend that the authors modify the title of the article and focus on the problem of cardioembolic stroke. The introduction needs a major revision. In this section, it is necessary to demonstrate the unresolved issues of the influence of seasonal fluctuations on the development of ischemic stroke in general and its cardioembolic subtype in particular. It is also necessary to add the purpose of this study.

Response: We have added more descriptions about the unsolved issues of the influence of seasonal fluctuations and the purpose of this study.

Point 2: Materials and methods: How was the sample size calculated? What are the inclusion and exclusion criteria? What are the criteria for the formation of observation groups? Which ethnic groups were included in this study? What are the characteristics of the seasons in Taiwan (average air temperature, average humidity, wind speed, etc.)?

Response: As described in [2.1 Study population], we retrospective reviewed all the acute ischemic stroke patients admitted to our stroke ward from January 2011 to December 2022. All patients were Taiwanese residents living near the hospital. The characteristics of the seasons were described in [2.4 Seasonal variation] and the average temperatures were described in [Results], both in manuscript text and in Figure 3, with a cited reference. We have also addsed average humidity and wind speed of Taipei city where the index hospital located in manuscript text. We have further added analyses of correlation of patient numbers with humidity and wind speed and have changed a new Figure 3A.

Point 3: Results: Add explanations of all the abbreviations that you used in Tables 1, 2.

Response: We have added explanation for LDL and TOAST subtypes in Tables 1 and 2.

Point 4: Discussion: What is the difference between the influence of the season on the incidence of cardioembolic stroke in Taiwan and other countries of Southeast Asia and other regions of the world?

Response: Most studies have focused on the impact of seasonal changes on the incidence of all types of ischemic stroke. Specific analysis of cardioembolic stroke is lacking. We have added more descriptions of meta-analysis studies of seasonal effects on stroke incidence in [Introduction].

Point 5: It is also necessary to update links to articles published more than 10 years ago.

Response: We have updated some cited references published more than 10 years ago.

Point 6: Comments on the Quality of English Language

A minor correction of the English language style is desirable.

Response: We have rechecked the English language style carefully.

Reviewer 2 Report

Comments and Suggestions for Authors

This paper reports on the results of a single centre 11-year study on clinical characteristics and outcomes of patients admitted for acute ischaemic stroke (AIS) with an emphasis in cadioembolic stroke (CES), and seasonal variations. The authors’ findings were consistent with the literature, including the higher proportion of cardioembolic stroke during winter compared to other stroke subtypes though this relatively novel finding is not well-described in the literature.

The authors may wish to attend to the following issues:

1. Title – as CES is a subtype of AIS, the authors may wish to tweak the title to: ‘Seasonal Variations and Comparison of Predictors of Unfavorable Outcomes Among Patients With Acute Ischemic Stroke and Those With Cardioembolic Stroke’

2. Abstract – there is no description of study Methods, subject demographics

3. Line 26 – I don’t think this study has shown that any of the information provided can ‘prevent CE’

4. Line 46 – it is not generally accepted that ‘any cardiac disorder’ causes CES – high-risk potential causes are accepted eg AF, mitral stenosis, mechanical heart valves, etc

5. Lines 46-56 – line spacing needs correction

6. line 56 – suggest to add ‘in particular’ after ‘patients’

Author Response

Response to Reviewer 2 Comments

Manuscript ID: Biomedicines-2755575

Title: Seasonal Variations and Comparison of Predictors of Unfavorable Outcomes Among Patients With Acute Ischemic Stroke and Those With Cardioembolic Stroke

Thanks to reviewer’s precious comments. We have checked the manuscript and have made essential revisions according to reviewer’s comments point-by-point. The revised portions in the manuscript were coded in red color.

Comments and Suggestions for Authors

This paper reports on the results of a single centre 11-year study on clinical characteristics and outcomes of patients admitted for acute ischaemic stroke (AIS) with an emphasis in cadioembolic stroke (CES), and seasonal variations. The authors’ findings were consistent with the literature, including the higher proportion of cardioembolic stroke during winter compared to other stroke subtypes though this relatively novel finding is not well-described in the literature.

The authors may wish to attend to the following issues:

Point 1. Title – as CES is a subtype of AIS, the authors may wish to tweak the title to: ‘Seasonal Variations and Comparison of Predictors of Unfavorable Outcomes Among Patients With Acute Ischemic Stroke and Those With Cardioembolic Stroke’

Response: We have changed the title as recommended above.

Pint 2. Abstract – there is no description of study Methods, subject demographics

Response: We have added more descriptions of study methods and subject demographics in [Abstract].

Point 3. Line 26 – I don’t think this study has shown that any of the information provided can ‘prevent CE’

Response: We have changed the description to “…may help develop stroke prevention strategies and facilitate early AF detection”.

Point 4. Line 46 – it is not generally accepted that ‘any cardiac disorder’ causes CES – high-risk potential causes are accepted eg AF, mitral stenosis, mechanical heart valves, etc

Response: We have changed the description to “The leading cause of cardioembolic stroke is atrial fibrillation”.

Point 5. Lines 46-56 – line spacing needs correction

Response: We have corrected the line spacing.

Point 6. line 56 – suggest to add ‘in particular’ after ‘patients’

Response: We have added ‘in particular’ after ‘patients’.

Reviewer 3 Report

Comments and Suggestions for Authors

The paper is a commendable effort in advancing the understanding of stroke outcomes and presents a wealth of data that could be of significant value to the medical community. By focusing on enhancing the interpretability of the results, clarifying methodological aspects, and providing a more nuanced discussion of its limitations, the paper could further its impact and contribution to the field. With these improvements, it has the potential to be an essential resource for researchers and clinicians alike.

Author Response

Response to Reviewer 3 Comments

Manuscript ID: Biomedicines-2755575

Title: Seasonal Variations and Comparison of Predictors of Unfavorable Outcomes Among Patients With Acute Ischemic Stroke and Those With Cardioembolic Stroke

Thanks to reviewer’s precious comments. We have checked the manuscript and have made essential revisions according to reviewer’s comments point-by-point. The revised portions in the manuscript were coded in red color.

Comments and Suggestions for Authors:

The paper is a commendable effort in advancing the understanding of stroke outcomes and presents a wealth of data that could be of significant value to the medical community. By focusing on enhancing the interpretability of the results, clarifying methodological aspects, and providing a more nuanced discussion of its limitations, the paper could further its impact and contribution to the field. With these improvements, it has the potential to be an essential resource for researchers and clinicians alike.

Response: Thanks to reviewer’s comments.

Reviewer 4 Report

Comments and Suggestions for Authors

In this study, we investigated seasonal and weekly changes in acute ischemic stroke (AIS) patients, including patients with EC stroke subtypes, and compared unfavorable outcome-related factors between all AIS patients and stroke patients. cerebrovascular with CE.

It is a cross-sectional study, which can be interesting due to the sample size and years of evolution, however, the statistics are very poor or poorly done.

add this citation: PMID: 31909910.

In Table 1, it is not clear how there are significant differences between such close values, such as heart rate, systolic pressure...

Likewise, how alcohol consumption has been measured

I do not approve Figure 1, although it is illustrative, it can be misleading. I do not understand what the criteria are for creating age groups, although age is a continuous variable, it provides more information treated as continuous.

Table 3 shows no significant differences in heart rate, with such a small difference.

The significant difference in the toast classification is not explained, and the post hoc is not pointed out either.

In table 4, I don't understand the correlation, what are the values that weekdays and holidays adopt?

Tables 6, where do you get those age values from?

I recommend making a decision tree that can give criteria to age cuts, and using more variables, it could be interesting.

In addition, with the year variable, you can see the evolution of the stroke by year, that is, does the age change each year or remains the same, and the rest of the characteristics to explain the difference between old and new strokes, and another study would be to know if there are differences in covid period.

Author Response

Response to Reviewer 4 Comments

Manuscript ID: Biomedicines-2755575

Title: Seasonal Variations and Comparison of Predictors of Unfavorable Outcomes Among Patients With Acute Ischemic Stroke and Those With Cardioembolic Stroke

Thanks to reviewer’s precious comments. We have checked the manuscript and have made essential revisions according to reviewer’s comments point-by-point. The revised portions in the manuscript were coded in red color.

Comments and Suggestions for Authors

In this study, we investigated seasonal and weekly changes in acute ischemic stroke (AIS) patients, including patients with EC stroke subtypes, and compared unfavorable outcome-related factors between all AIS patients and stroke patients. cerebrovascular with CE.

It is a cross-sectional study, which can be interesting due to the sample size and years of evolution, however, the statistics are very poor or poorly done.

Point 1: add this citation: PMID: 31909910.

Response: We have added this citation in the section of [Limitations].

Point 2: In Table 1, it is not clear how there are significant differences between such close values, such as heart rate, systolic pressure...

Response: Given the relatively large sample size in this study, some close values may exhibit significant difference between two groups. Because most of the variables were not normally distributed, we selected non-parametric analysis to present all the results. The mean values of SBP in men and in women were 163±31 mmHg and 166±33 mmHg, respectively (two-sample t-test p = 0.015). The mean values of HR in men and in women were 81±17 BPM and 83±18 BPM, respectively (two-sample t-test p = 0.002). Both of SBP and HR exhibited statistically significant through parametric analyses as well. Some variables may show significant differences in univariate analysis, however, such differences may not demonstrate relevant clinical importance and may become nonsignificant in multivariable analysis. 

Point 3: Likewise, how alcohol consumption has been measured

Response: Alcohol consumption is defined as habitual consumption of alcoholic beverage at least once a week in the past year. We have added the definition of alcohol consumption in the text: “alcohol consumption (defined as habitual consumption of alcoholic beverage at least once a week in the past year)”.

Point 3: I do not approve Figure 1, although it is illustrative, it can be misleading. I do not understand what the criteria are for creating age groups, although age is a continuous variable, it provides more information treated as continuous.

Response: Either continuous or categorial variables is suitable for demonstrating the trend of aging effect. Table 2 reports age as a continuous variable, showing that patients with cardioembolism had the highest median age. To better demonstrate prevalence differences among different age groups, we stratified age into four groups. The age categories are modified based on a departmental standard (age categories, life cycle grouping) from Statistics Canada, where senior adults are defined as 65 years and over with each segmentation age of 5 years old. (https://www.statcan.gc.ca/en/concepts/definitions/age2).     

Point 3: Table 3 shows no significant differences in heart rate, with such a small difference.

Response: This is the same situation as mentioned in Point 2.

Point 4: The significant difference in the toast classification is not explained, and the post hoc is not pointed out either.

Response: Some descriptions of the significant differences in the TOAST classification have been distributed in the discussion of each section.

Point 5: In table 4, I don't understand the correlation, what are the values that weekdays and holidays adopt?

Response: We have changed the title of Table 4 into “Comparison of clinical features between weekdays and holidays in terms of stroke onset day in 4,040 patients with acute ischemic stroke”. The definitions of weekdays and holidays are described in “2.4 Seasonal Variation and Weekday Identification”.

Point 6: Tables 6, where do you get those age values from?

Response: All the cutoff values for continuous variables in Tables 5 and 6 are described in the text: “we converted significant continuous variables into dichotomous variables by using the optimal cutoff value determined according to the Youden index for comparing all patients with AIS and patients with CE stroke.” The statistical methods are described in [Methods]: “The continuous variables of age, initial NIHSS score, and laboratory data were converted into dichotomous variables by using the optimal cutoff value determined according to the Youden index with a receiver operating characteristic (ROC) curve plotted for unfavorable outcome and death prediction.”

 Point 7: I recommend making a decision tree that can give criteria to age cuts, and using more variables, it could be interesting.

Response: Thanks for reviewer’s recommendation. Considering that decision tree is a decision support hierarchical model commonly used in operations research, and may not be an appropriate method in our study, we decided to use nomograms comprising both continuous and categorial variables for outcome prediction. We have developed nomograms for predication of death among all patients with AIS and patients with cardiombolic stroke.

Point 8: In addition, with the year variable, you can see the evolution of the stroke by year, that is, does the age change each year or remains the same, and the rest of the characteristics to explain the difference between old and new strokes, and another study would be to know if there are differences in covid period.

Response: Given that the study period was only 12 years, most variables did not exhibit yearly secular trend. Only the prevalence of cardioembolism exhibits a trend of yearly increment and the rate of in-hospital complication exhibits a trend of yearly decrement. It is true that in 2020, the height of the COVID-19 pandemic period, the number of patients admitted to hospital (n = 266) was well below the annual average (n = 337). We have added such findings in [Results] section.

Round 2

Reviewer 1 Report

Comments and Suggestions for Authors

I thank the authors for answering my questions and modifying the manuscript.

Reviewer 4 Report

Comments and Suggestions for Authors

The authors have made the suggested changes.